# Ball-Milling Preparation of La$^{3+}$/TiO$_2$ Photocatalyst and Application in Photodegradation of PVC Plastics

**Yan Zhang** [1,2], **Tianyi Sun** [1,2,*], **Dashuai Zhang** [1,2], **Chen Li** [1,2], **Jinrui Liu** [1,2], **Bangsen Li** [1,2] and **Zaifeng Shi** [1,2,*]

1   Key Laboratory of Water Pollution Treatment & Resource Reuse, Hainan Normal University, Haikou 571158, China
2   College of Chemistry and Chemical Engineering, Hainan Normal University, Haikou 571158, China
*   Correspondence: tianyi870328@163.com (T.S.); zaifengshi@163.com (Z.S.)

**Abstract:** The fact that the use of a large number of plastic products has brought serious pollution to the environment has always been the focus of global attention. The development of photocatalytic degradable plastics is one of the effective ways to solve the problem of "white pollution". In this work, La$^{3+}$ modified TiO$_2$ nanoparticles were prepared by ball milling and characterized. La$^{3+}$/TiO$_2$ was mixed with Polyvinyl chloride (PVC) plastic to make a photodegradable composite film, and the photodegradation performance and mechanical properties of films were evaluated. The photodegradable films were characterized by infrared spectroscopy (FT-IR), X-ray diffractometry (XRD), and scanning electron microscopy (SEM). After 30 h UV irradiation, the weight loss rate of the PVC was only 2.12%, while that of the TiO$_2$/PVC reached 8.94%. The accelerating of the degradation rate was due to the mixing of TiO$_2$ into PVC. As for the La$^{3+}$/TiO$_2$/PVC composite film, when the mass percentage of La$^{3+}$/TiO$_2$ was 1.5%, the weight loss rate of La$^{3+}$/TiO$_2$/PVC sample reached a maximum of 17.78%, which was eight times the degradation rate of PVC and two times the degradation rate of TiO$_2$/PVC. The La$^{3+}$/TiO$_2$/PVC film showed good photodegradability. La is a transition metal element with a special 4f electronic structure. The reaction mechanism of photodegradation of PVC by the interaction of La$^{3+}$ and TiO$_2$ were discussed.

**Keywords:** La$^{3+}$ modified TiO$_2$; La$^{3+}$/TiO$_2$/PVC composite film; ball milling method; photodegradation

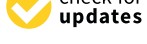



## 1. Introduction

PVC is one of the five biggest general-purpose plastics, having the characteristics of non-combustibility, corrosion resistance, chemical resistance, insulation, and good mechanical properties. Recently, PVC was widely used in industry, products, daily necessities, building materials, environmental protection materials, etc. [1,2]. However, with the increasing use of one-off plastic products, the pollution resulting from the plastic waste has become more and more serious [3,4]. Plastic pollution has always been a hot spot of global concern. The treatment methods are incineration, landfill [5], recycling, chemical degradation, and the production of degradable plastics [6,7]. Photodegradable plastics have been widely studied, which is an ideal way to solve the "white pollution" [8,9].

Photocatalysis is a green-friendly technology that combines abundant solar energy resources with environmental cleanup and resource reuse and has received widespread attention [10]. TiO$_2$, ZnO, CdS, and Fe$_2$O$_3$ are common semiconductor materials [11]. Because of its low cost, non-toxicity and its photocatalytic activity for the decomposition of various environmental pollutants, TiO$_2$ is widely used in the field of photodegradable plastics [12,13]. At the present stage, the main modification method of TiO$_2$ is to regulate the structure and composition of the catalyst, including the regulation of TiO$_2$ crystal structure and defects, grain size, and energy band position. TiO$_2$ can be doped with metal, non-metal, precious metal, and other elements, and can also be modified by surface photosensitization and semiconductor composite [14,15]. The modification of TiO$_2$ is aimed

at expanding the absorption range of $TiO_2$ to sunlight, promoting the efficiency of charge separation, inhibiting the recombination of photo-generated carriers and improving the stability of catalyst [16]. Rare earth is a type of transition metal element with a special 4f electronic structure. Lanthanum is second only to cerium in rare earth elements and is rich in resources. Therefore, rare earth doped modified $TiO_2$ has been widely studied to improve photocatalytic activity [17–20].

Due to its simple operation and low cost, the ball milling method is an effective method in the industry to efficiently mix solids and solid reactants by using the rotation or vibration of a ball mill [21]. The rare earth modified $TiO_2$ photocatalyst made by the ball milling method applied to the degradation of PVC waste plastic is of great significance for solving the problem of environmental pollution from the viewpoint of practical application.

In this paper, $La^{3+}/TiO_2$ photocatalyst was prepared by the ball milling method. The $La^{3+}/TiO_2/PVC$ composite film was prepared by introducing $La^{3+}/TiO_2$ into PVC plastic. After being compared with the PVC and $TiO_2/PVC$ films, the $La^{3+}/TiO_2/PVC$ film was characterized and its photodegradability was also studied. Moreover, the reaction mechanisms of the photodegradation of PVC by the interaction of $La^{3+}$ and $TiO_2$ were discussed.

## 2. Experimental

### 2.1. Materials

$TiO_2$ (Analytical reagent, the average particle size = 19.8 nm) was purchased from Komiou Chemical Reagent Co., Ltd., Tianjin, China. The PVC polymer was produced by Aladdin (K-Value: 62–60). $La_2O_3$ was provided by Hunan Rare Earth Metal Materials Research Institute. N,N-dimethylformamide (AR), abbreviated as DMF, was produced by the Damao Reagent Company, Tianjin, China. Hydrochloric acid (HCl, 36.0–38.0%) was provided by Fusheng Industry Co., Ltd., Shanghai, China.

### 2.2. Characterization

Using Shimadzu UV-2600 ultraviolet-visible diffuse reflectance instrument, with $BaSO_4$ as a reference, the ultraviolet-visible absorption spectrum of the films was measured. Under the condition of vacuum $5 \times 10^{-4}$ Pa, a JEOL Ltd. JSM-7100F thermal field emission scanning electron microscope (SEM, Electronics Co., Ltd., Japan) was used to test the surface morphology of the films, and the voltage was 5.0 kV. The samples were characterized by Thermo Scientific Nicolet 6700 Fourier transform infrared spectroscopy (FT-IR, Thermo Fisher Instruments Co., Waltham, MA, USA). The crystal phase of the materials was detected on an X-ray diffractometer (XRD, Pruck instruments Co., Germany) with Cu/K radiation, and the diffraction intensity was recorded in the 2θ range of 10–80° at a scanning speed of 5°/min. The digital display tensile testing machine was used to test the mechanical properties of the film, and purchased from Guangdong Zhongye Jingke Instrument Equipment Co., Ltd., Guangdong, China. The film was cut into a rectangle of 50 mm × 10 mm and fixed on the digital tensile testing machine with an initial spacing of 20 mm, running at a 50 mm/min. The thickness of the film was measured 5 times with the NSCING electronic digital micrometer, and the average value was taken.

### 2.3. Preparation of the $La^{3+}/TiO_2$ Photocatalyst

The $La^{3+}/TiO_2$ photocatalyst was synthesized by a ball milling method, according to the following procedure: Firstly, 5 g of $TiO_2$, six Φ 10 and twenty Φ 6 agate balls were added to the agate jar. After being ground for several minutes, $La_2O_3$ with the corresponding mass percentage of 0.6% and a small amount of deionized water were added and heated to 80 °C. Then, the hydrochloric acid with the volume ratio of 1:1 was slowly added into the above solution and stirred until the solid was dissolved completely. Then, the deionized water was added to the total volume of 10 mL. The speed of the ball mill was set to 500 r/min, and the ball milling time was 4 h. The ball-milled slurry was washed with water, centrifuged, dried, and ground to obtain the $La^{3+}/TiO_2$ photocatalyst.

### 2.4. Preparation of Film and Photocatalytic Experiments

The preparation process and photocatalysis experiment of the $La^{3+}/TiO_2/PVC$ film are shown in Figure 1. All raw materials need to be vacuum dried. The doping content of the $La^{3+}/TiO_2$ photocatalyst in the film were as a mass percentage of 0.1, 0.5, 1, 1.5, and 2, respectively. Hence, 2 g of PVC and 15 g of organic solvent DMF were added separately, heated, and stirred for 1 h. The mixed solution was ultrasonicated for 10 min so that the photocatalyst could be better mixed with the polymer. At room temperature, the mixed solution was dropped on the automatic film scraping machine and run at constant speed to prepare a film with a thickness of 150 μm. The flat plate was removed from the automatic film scraping machine and immersed in deionized water to obtain a composite film. The composite film was placed in the XPA-System photochemical reactor for light irradiation. The light source was a 300 W medium-pressure UV lamp with main wavelength of 365 nm. and the distance between the light source and the sample was 20 cm. The sample was taken out every 5 h, and the total illumination time was 30 h. The sample was taken out for storage and performance testing. The weight loss calculation formula is presented in Equation (1):

$$Weightloss\ (\%) = W_0 - W_t/W_0 \times 100\% \tag{1}$$

where $W_t$ is the weight of film at after illumination and $W_0$ is the initial weight of film.

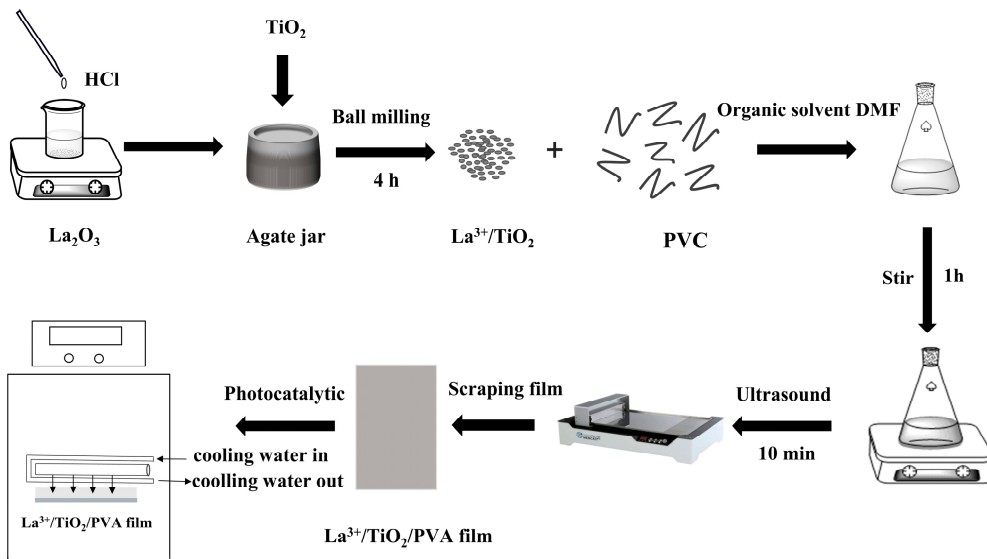

**Figure 1.** Preparation process of $La^{3+}/TiO_2/PVC$ composite film by using $La^{3+}/TiO_2$ photocatalyst and the schematic diagram of the photodegradation equipment.

## 3. Results and Discussion

### 3.1. Characterization of $La^{3+}/TiO_2$ Photocatalyst

Figure 2A shows the XRD spectrum of $La^{3+}/TiO_2$ and raw $TiO_2$ after ball milling. Compared with $TiO_2$, the 3 mol% $La^{3+}/TiO_2$ still maintained the anatase crystal after doping, and no other peaks of rutile crystal was observed. This suggested that $La^{3+}$ doped $TiO_2$ could inhibit the appearance of rutile crystal by ball milling method [22]. Moreover, no obvious characteristic peaks of the $La^{3+}$ were found, possibly due the small content of $La^{3+}$ doped into the $TiO_2$ lattice. Furthermore, the ionic radius of $La^{3+}$ was larger than that of $Ti^{4+}$, so these doped ions might have difficulty in entering the $TiO_2$ lattice. The peak position of the (101) crystal plane in $La^{3+}/TiO_2$ appeared red-shifted to 25.4°, indicating that $La^{3+}$ could be highly dispersed on the $TiO_2$ surface in a free state [23]. As shown in Figure 2B, the EDX spectra indicate that the sample was mainly formed of Ti, O, and La, suggesting that $La^{3+}$ successfully doped to $TiO_2$. $La^{3+}$ was dispersed on the surface of

$TiO_2$, and it had a squeezing effect on the $TiO_2$ lattice, resulting in more crystal defects and distortions, which had a certain effect on the $TiO_2$ crystal [24,25].

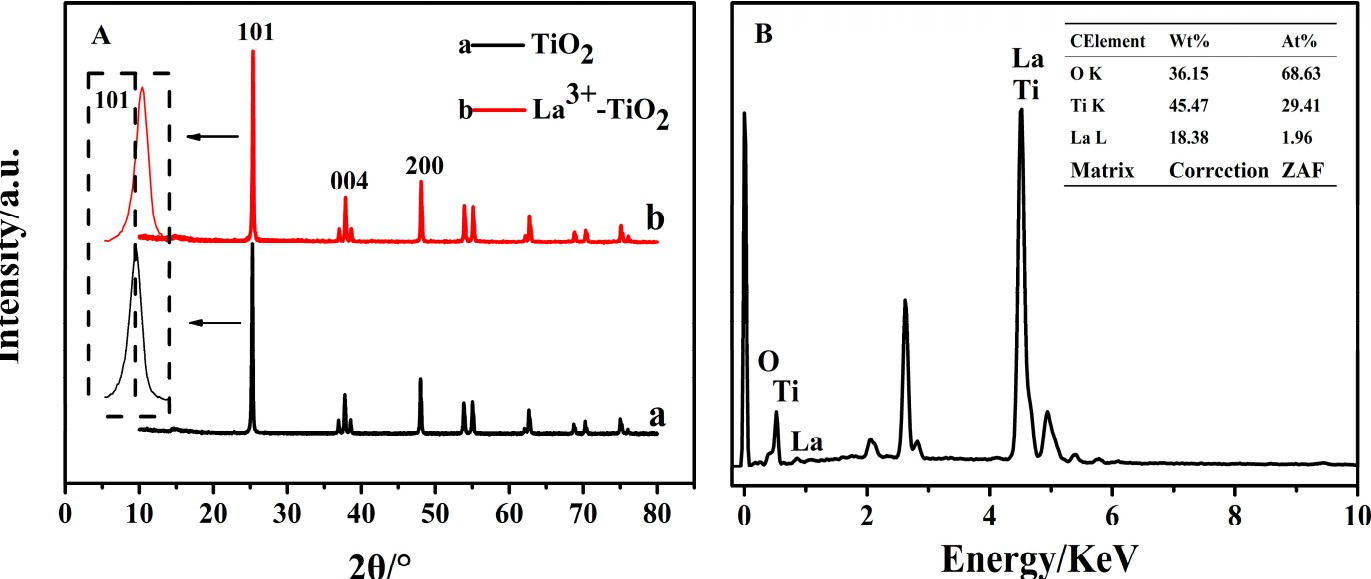

**Figure 2.** Characterization of the $La^{3+}/TiO_2$. (**A**) XRD pattern; (**B**) EDX spectra and component molar ratio.

Figure 3 shows the FT-IR spectrum of raw $TiO_2$ and 0.6 wt% $La^{3+}/TiO_2$. A strong characteristic peak between 2700 and 3600 $cm^{-1}$ was attributed to -OH in the $H_2O$ molecule, while the corresponding bending vibration peak appeared near 1630 $cm^{-1}$ [26]. The characteristic peaks between approximately 500 and 900 $cm^{-1}$ belonged to the anatase O-Ti-O bond [27]. After $La^{3+}$ doping, the strong peaks of the defective -OH groups obviously weakened, as the OH groups on the surface of $TiO_2$ could be covalently bonded to $La^{3+}$ [28]. A decrease in the characteristic peak of the anatase O-Ti-O bond was attributed to the covalent band between $La^{3+}$ and $TiO_2$ particles, further indicating that $La^{3+}$ was successfully doped to $TiO_2$.

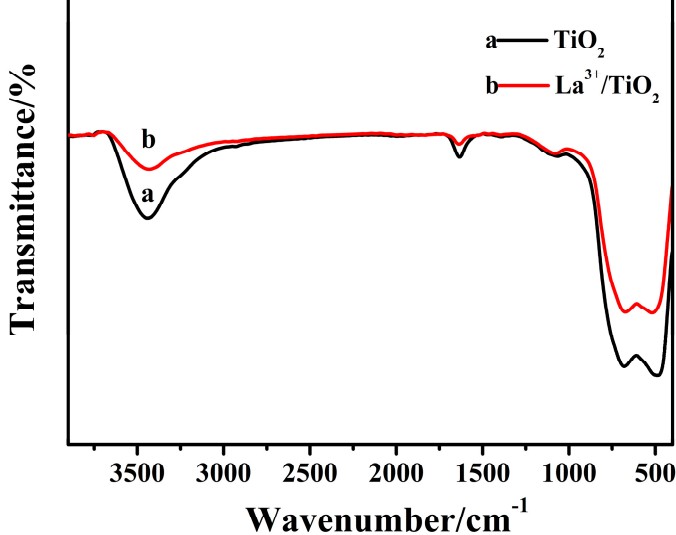

**Figure 3.** The FT-IR spectra of $TiO_2$ and $La^{3+}/TiO_2$.

Figure 4A shows the UV-visible absorption spectra of $TiO_2$ and $La^{3+}/TiO_2$. Compared with $TiO_2$, the UV absorption of $La^{3+}/TiO_2$ prepared by the ball milling method had little

change except a slight redshift. However, the absorption of $La^{3+}/TiO_2$ slightly enhanced in the visible region. The red shift of the absorption band was one of the important factors that made the sample have good photocatalytic performance under visible light. The reduced band gap energy of $La^{3+}/TiO_2$ could excite and produce more photo-generated electron-hole pairs, thus effectively improving the photocatalytic efficiency [29]. As shown in Figure 4B, the light absorption properties of PVC, $TiO_2/PVC$, $La^{3+}/TiO_2/PVC$ composite films were also investigated. The PVC film had little absorption in both ultraviolet and visible region. The absorption of the $TiO_2/PVC$ film significantly enhanced in the near ultraviolet region. The absorption band of $La^{3+}/TiO_2/PVC$ appeared red-shifted and could absorb light energy in the visible region. The red shift was mainly because part of $La^{3+}$ was embedded on $TiO_2$ to form defect energy levels, which reduced the energy level spacing during electron transition [30]. Therefore, it was of great significance to use $La^{3+}/TiO_2$ with the higher photocatalytic activity to enhance the degradation performance of PVC.

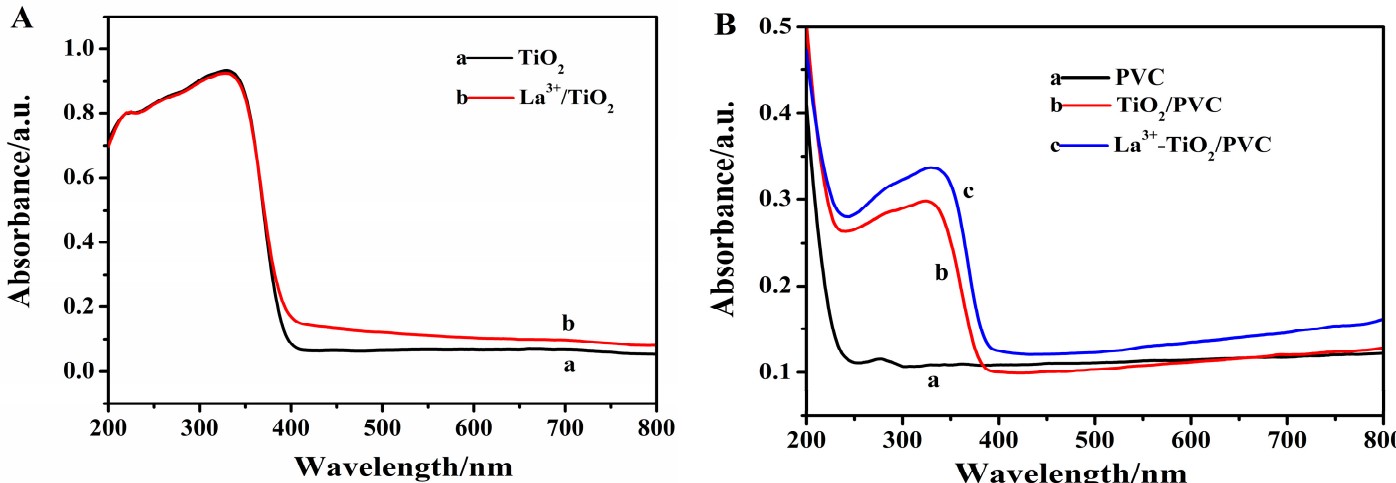

**Figure 4.** UV-visible absorption spectrum. (**A**) (a) $TiO_2$ (b) $La^{3+}/TiO_2$; (**B**) (a) Pure PVC sample, (b) $TiO_2/PVC$ sample, (c) $La^{3+}/TiO_2/PVC$ sample.

*3.2. Photodegradation of $La^{3+}/TiO_2/PVC$*

In order to obtain a film with the best degradation effect, $La^{3+}/TiO_2$ was mixed with PVC to make the composite films with different mass percentages, and the photodegradation rates of the above $La^{3+}/TiO_2/PVC$ composite films were compared with those of PVC and $TiO_2/PVC$ after irradiation (Figure 5). After light irradiation, the weight loss rate of the PVC was only 2.12%, while that of the $TiO_2/PVC$ reached 8.94%. The accelerating of the degradation rate was due to the mixing of $TiO_2$ into PVC. As for the $La^{3+}/TiO_2/PVC$ composite film, when the mass percentage of $La^{3+}/TiO_2$ was 1.5%, the weight loss rate of $La^{3+}/TiO_2/PVC$ sample reached a maximum of 17.78%, which was eight times the degradation rate of PVC and two times the degradation rate of $TiO_2/PVC$. Obviously, the photocatalytic activity of $La^{3+}/TiO_2/PVC$ film increased greatly. Rare earth $La^{3+}$ modified $TiO_2$ could inhibit the recombination of photo-generated electrons and holes and accelerate the formation of active oxides [31]. Moreover, the ball milling method was beneficial to the uniform distribution of $La^{3+}$ and suppressed the $TiO_2$ agglomeration. After repeated grinding, the $TiO_2$ particles could be made smaller and its surface area increased [32,33], which supplied more photocatalytic active sites and further accelerated the degradation rate of PVC.

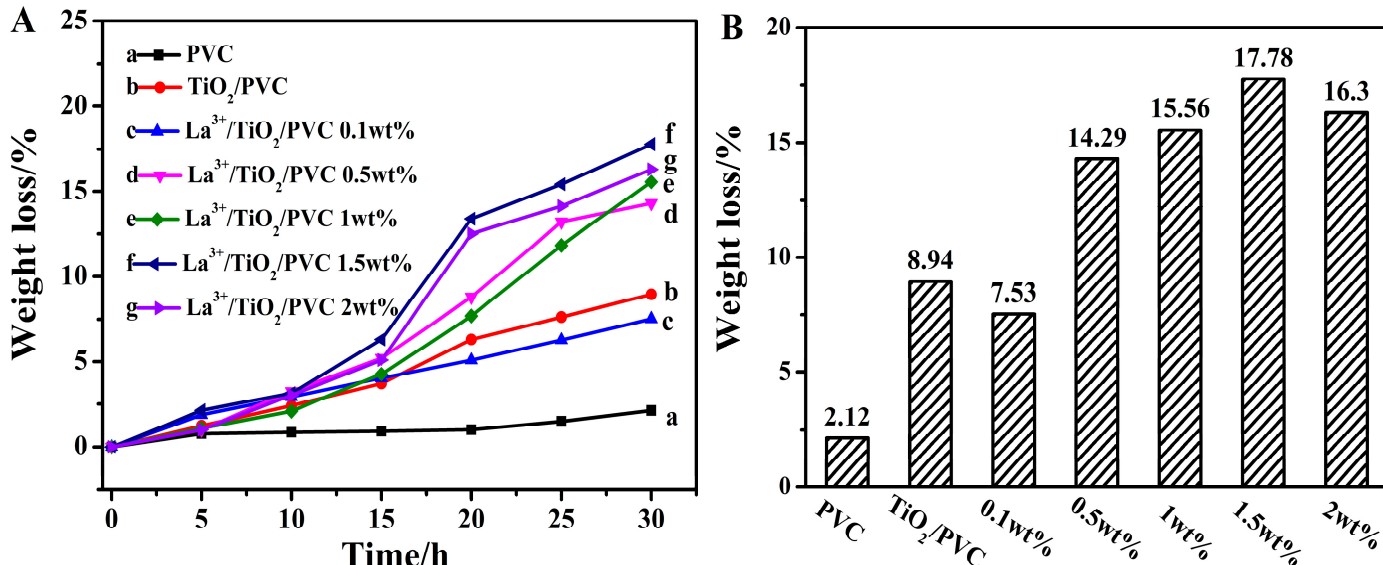

**Figure 5.** (**A**) Curves of weight loss of degradation process. (a) Pure PVC sample, (b) $TiO_2$/PVC sample, (c) $La^{3+}$/$TiO_2$/PVC ($La^{3+}$/$TiO_2$ 0.1 wt%), (d) $La^{3+}$/$TiO_2$/PVC (0.5 wt%), (e) $La^{3+}$/$TiO_2$/PVC (1 wt%), (f) $La^{3+}$/$TiO_2$/PVC (1.5 wt%), (g) $La^{3+}$/$TiO_2$/PVC (2 wt%). (**B**) Maximum weight loss rate.

Figure 6A compares the FT-IR spectrum of pure PVC, $TiO_2$/PVC, $La^{3+}$/$TiO_2$/PVC samples after being UV irradiated for 30 h with that of the original samples. As shown in Table 1, The characteristic peaks of C-H for the original PVC film appeared at 2973 $cm^{-1}$, 2917 $cm^{-1}$, and 1427 $cm^{-1}$ [34]. The characteristic peaks at 698 $cm^{-1}$ and 624 $cm^{-1}$ belonged to the C-Cl, and the characteristic peaks at 1097 $cm^{-1}$ belonged to the C-C of the PVC film [35]. Moreover, characteristics peak of $TiO_2$ could be seen in the spectrum of $TiO_2$/PVC and $La^{3+}$/$TiO_2$/PVC samples. The stretching vibration peak at 478 $cm^{-1}$ belonged to Ti-O-Ti [36]. Obviously, a new band of at 1720 $cm^{-1}$ was assigned to the C = O structure after illumination. The intensity of the characteristic peak of the C = O structure of the $La^{3+}$/$TiO_2$/PVC film increased gradually after 0 h,10 h, 20 h, and 30 h of irradiation (Figure 6B), indicating that the film was degraded with the continuous irradiation of light source. Due to the fact that $La^{3+}$/$TiO_2$ produces active oxides under UV irradiation. $O_2^-$ or OH free radicals attacked the PVC molecular chain, and $O_2$ also participated in the reaction to generate oxygen-containing groups and caused the photodegradation of PVC [37]. $La^{3+}$ can effectively separate the hole-electron pair of $TiO_2$ to generate more free radicals to attack PVC, which can increase the degradation rate of plastics.

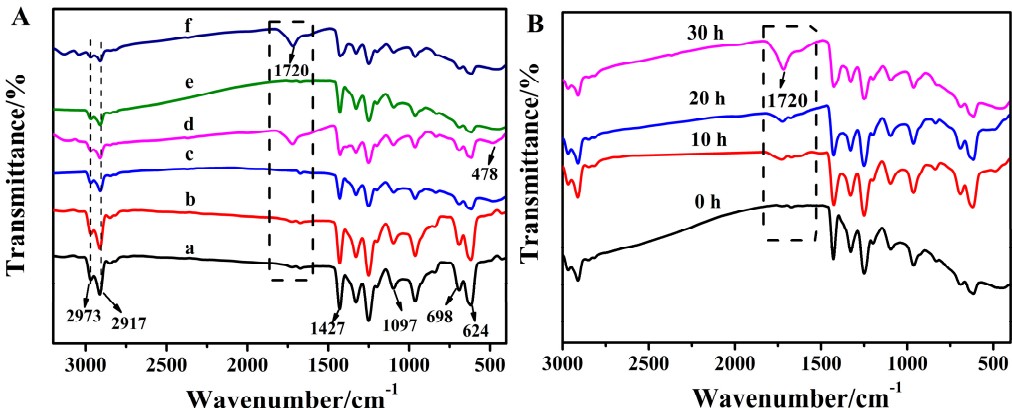

**Figure 6.** FT-IR spectra of polymer films. (**A**) (a) PVC before irradiation (b) PVC after irradiation, (c) $TiO_2$/PVC before irradiation (d) $TiO_2$/PVC after irradiation, (e) $La^{3+}$/$TiO_2$/PVC before irradiation (f) $La^{3+}$/$TiO_2$/PVC after irradiation. (**B**) $La^{3+}$/$TiO_2$/PVC after irradiation for 0 h, 10 h, 20 h, 30 h.

**Table 1.** The characteristic peak of infrared spectrum structure.

| Serial Number | Wavenumber/cm$^{-1}$ | Characteristic Peak |
|:---:|:---:|:---:|
| 1 | 2973, 2917, 1427 | C-H |
| 2 | 698, 624 | C-Cl |
| 3 | 1097 | C-C |
| 4 | 478 | Ti-O-Ti |
| 5 | 1720 | C=O |

Figure 7 shows the XRD patterns of PVC, TiO$_2$/PVC, and La$^{3+}$/TiO$_2$/PVC films before and after UV irradiation. There was no obvious sharp characteristic diffraction peak of PVC, which clearly illustrated that PVC polymer was an amorphous structure, and the peak distribution was wide and low [38]. The characteristic diffraction peaks of TiO$_2$ appeared in the TiO$_2$/PVC film. After light illumination, the peaks of TiO$_2$/PVC and La$^{3+}$/TiO$_2$/PVC films disappeared at 13°~20°, while the pure PVC did not change. This was due to the destruction of the PVC structure in the photocatalyst/UV system. A new peak appeared at 32.5° in all composite films after light illumination, because several new small molecules appeared in the process of breaking PVC, which corresponded to the FT-IR spectrum.

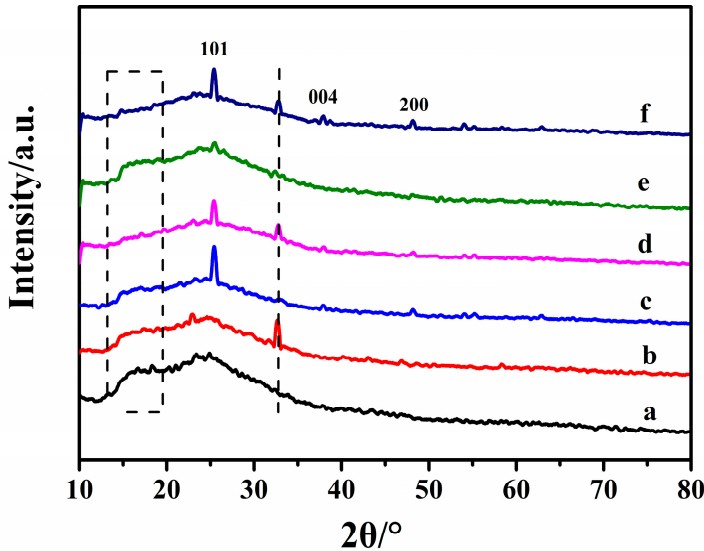

**Figure 7.** The XRD pattern for films. (**a**) PVC before irradiation (**b**) PVC after irradiation, (**c**) TiO$_2$/PVC before irradiation (**d**) TiO$_2$/PVC after irradiation, (**e**) La$^{3+}$/TiO$_2$/PVC before irradiation (**f**) La$^{3+}$/TiO$_2$/PVC after irradiation.

The surface structure of the film could be clearly observed by SEM. As shown in Figure 8, the surface of pure PVC, TiO$_2$/PVC and La$^{3+}$/TiO$_2$/PVC film had a certain size of hole before illumination. After UV irradiation, the size of the holes in PVC film increased slightly, while that in the TiO$_2$/PVC film significantly increased in terms of porosity and appeared to be crispy, indicating that a photo-oxidation reaction occurred on the film surface. The La$^{3+}$/TiO$_2$/PVC composite film could be seen to gradually undergo photooxidation after 10 h, 20 h, and 30 h UV exposure. Under 30 h illumination, the hole diameter increased to approximately 2–8 μm, and the La$^{3+}$/TiO$_2$ particles could be observed on the La$^{3+}$/TiO$_2$/PVC sample (Figure 8C4). As shown in the inset in Figure 8C4, the film appeared to fracture and the lanthanum ion-doped TiO$_2$ photocatalyst was observed at the place of the break, which indicated that the photo-oxidation was caused by the exposed La$^{3+}$/TiO$_2$. Therefore, the photodegradable film prepared by this method had certain application prospects for environmental friendliness.

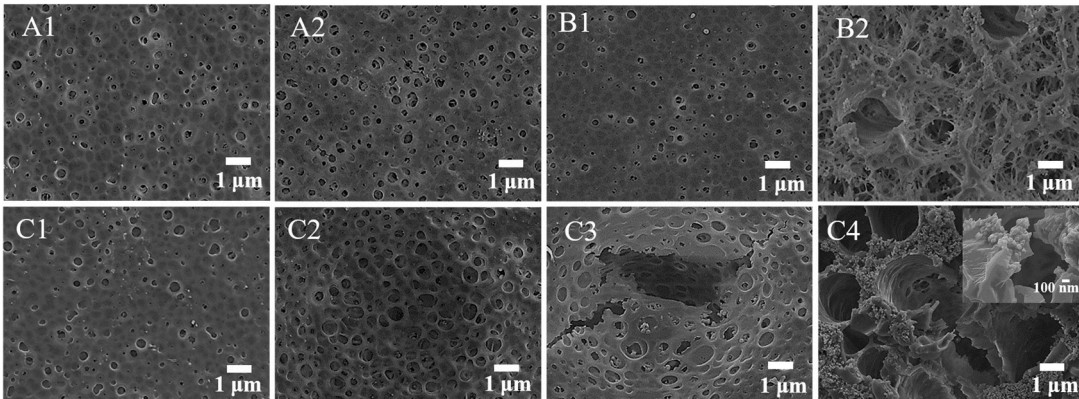

**Figure 8.** SEM images of films. (**A1**) PVC before irradiation, (**A2**) PVC after irradiation; (**B1**) $TiO_2$/PVC before irradiation, (**B2**) $TiO_2$/PVC after irradiation; (**C1**) $La^{3+}$/$TiO_2$/PVC before irradiation, $La^{3+}$/$TiO_2$/PVC after irradiation for (**C2**) 10 h, (**C3**) 20 h, (**C4**) 30 h.

### 3.3. The Performance Characterization of La³⁺/TiO₂/PVC

One of the important reasons why the additive degradable plastics have not been widely used was the poor mechanical properties of plastics. Therefore, it is of great significance to study the tensile properties of the $La^{3+}$/$TiO_2$/PVC plastics. There were two important factors influencing the mechanical properties of materials: (1) the shape, orientation, and distribution of the additive, (2) the interaction between the additive and the matrix [39,40]. As shown in Table 2, after mixing PVC and $TiO_2$ or $La^{3+}$/$TiO_2$, the tensile strength and elongation of PVC decreased slightly. The slight decrease in mechanical properties might be due to the mixture of amorphous and crystalline $TiO_2$ in the PVC chain, which reduced the polymer interaction between chains [41]. However, the $La^{3+}$/$TiO_2$/PVC degradable was made by this method and belonged to an environment-friendly composite material.

**Table 2.** Mechanical properties of these kinds of different films.

| Sample | Thickness (mm) | Tensile Strength (MPa) | Elongation at Break (%) |
|---|---|---|---|
| PVC | $0.12 \pm 0.01$ | $2.10 \pm 0.20$ | $31 \pm 0.08$ |
| $TiO_2$/PVC ($TiO_2$ 1 wt%) | $0.11 \pm 0.01$ | $1.70 \pm 0.10$ | $26 \pm 0.10$ |
| $La^{3+}$/$TiO_2$/PVC ($La^{3+}$/$TiO_2$ 1.5 wt%) | $0.13 \pm 0.01$ | $1.71 \pm 0.17$ | $31 \pm 0.04$ |

The thermal degradation of PVC was mainly divided into two stages (Figure 9). The first stage was the removal of HCl from PVC, where most of the degradation products in the second stage were cyclic compounds [42]. Compared with pure PVC film, the initial degradation temperature of PVC composite films had no obvious change. However, the overall degradation temperature shifted to a higher temperature, and the thermal stability of PVC composite film was improved. The mass of films did not decrease around 100 °C, indicating that the films did not contain water molecules. At 800 °C, the PVC achieved complete thermal degradation.

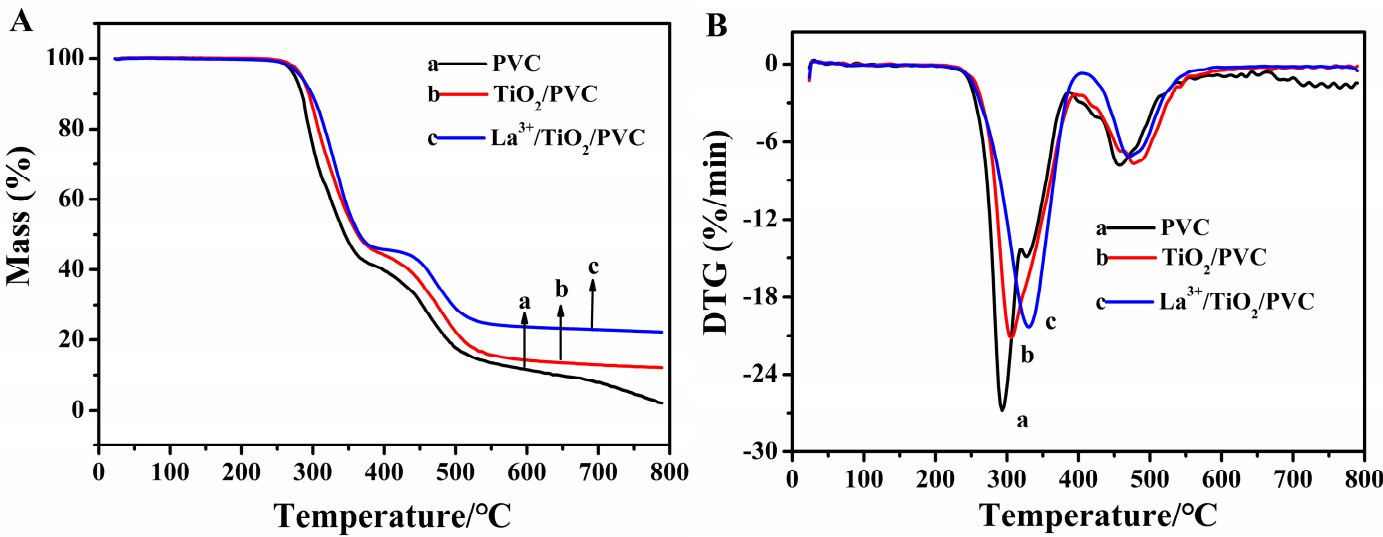

**Figure 9.** (**A**) TG and (**B**) DTG curves of three kinds of films.

### 3.4. Mechanism Discussion

$La^{3+}$ doping modified $TiO_2$ has important significance for the increase of the $TiO_2$ photocatalytic activity. The reaction mechanism is shown in Figure 10. $La^{3+}$ doped $TiO_2$ could inhibit the growth of $TiO_2$ grains and cause lattice distortion by the ball milling process. The grain refinement increased the specific surface area of $TiO_2$, provided more photocatalytic active sites, and improved the photocatalytic activity of $TiO_2$. When La ions were doped into the $TiO_2$ lattice, $La^{3+}$ could capture photo-generated electrons to form $La^{2+}$ (Formula (2)), and $La^{2+}$ could easily form superoxide radicals with $O_2$ (Formula (3)), effectively inhibiting photo-generated carrier recombination (Formula (4)). Under the sufficient light energy, the valence band electrons of $TiO_2$ were excited to the conduction band. The outer electronic structure of the La element contained only one electron in the d orbit and the completely empty f orbit. Therefore, the empty orbits provided the transfer orbits for electrons (Formula (5)), which could be used as shallow traps to capture the photogenerated electrons, as well as facilitate the separation of photogenerated electron hole pairs. OH was formed by the reaction of photogenerated holes with $H_2O$ on $TiO_2$ (Formula (6)). The photogenerated electrons reacted with $O_2$ to produce $O_2^{-}$ and OH (Formulas (7)–(10)) [43]. During the photodegradation process of high-molecular polymers, the PVC molecular chains were attacked by $O_2^{-}$ and OH to generate a free radical chain. According to the chemical bond energies of C-Cl, C-C, and C-H, 328 kJ/mol, 348 kJ/mol, and 413 kJ/mol, respectively, C-Cl was more easily destroyed by the free radicals generated under UV light (Formulas (11) and (12)). Then, the free radical chain continued to react with $O_2$ to form oxygen-containing compounds, which were decomposed into small molecules under the action of UV light, and finally completely mineralized (Formulas (13) and (14)). Moreover, $La^{3+}$ doping modified $TiO_2$ could inhibit the growth of $TiO_2$ grains and cause lattice distortion by ball milling process. The grain refinement increased the specific surface area of $TiO_2$, supplying more photocatalytic active sites and improving the photocatalytic activity of $TiO_2$. The $La^{3+}$ doping would also change the band gap of $TiO_2$, so that $La^{3+}/TiO_2$ had absorption in the visible region [44]. Therefore, it is of great significance to use $La^{3+}/TiO_2$ with the higher photocatalytic activity to enhance the degradation performance of PVC.

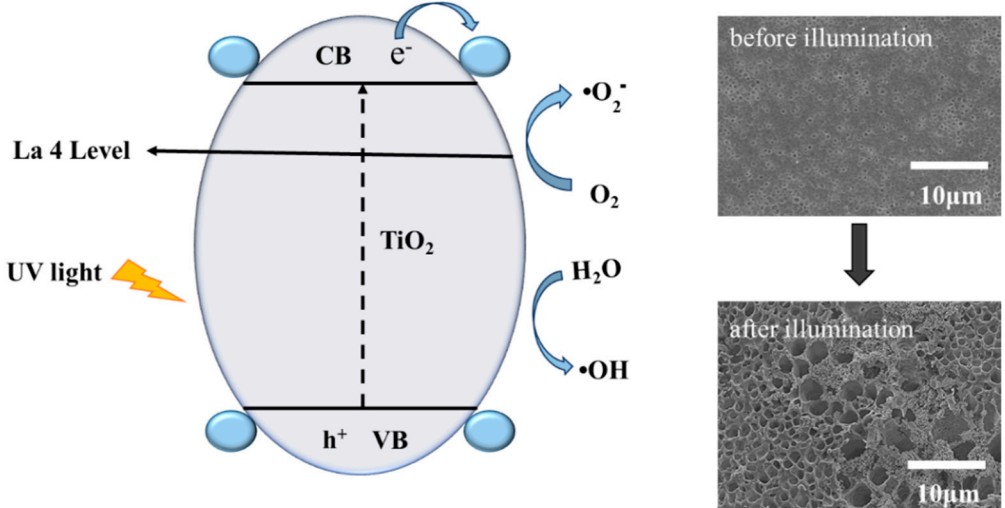

**Figure 10.** Photocatalytic degradation mechanism of $La^{3+}/TiO_2/PVC$.

$$La^{3+} + e^- \rightarrow La^{2+} \qquad (2)$$

$$La^{2+} + O_2 \rightarrow La^{3+} + \bullet O_2{}^- \qquad (3)$$

$$2 \bullet O_2{}^- + 2 H^+ + e^- \rightarrow \bullet O_2H + OH^- + O_2 \qquad (4)$$

$$TiO_2 + h\nu \rightarrow TiO_2 (e^- + h^+) \qquad (5)$$

$$h^+ + H_2O \rightarrow \bullet OH \qquad (6)$$

$$e^- + O_2 \rightarrow \bullet O_2{}^- \qquad (7)$$

$$\bullet O_2{}^- + H_2O \rightarrow \bullet O_2H + OH^- \qquad (8)$$

$$2 \bullet O_2H \rightarrow H_2O_2 + O_2 \qquad (9)$$

$$H_2O_2 + h\nu \rightarrow 2 \bullet OH \qquad (10)$$

$$-(CH_2CHClCH_2CHCl)- + \cdot OH + h\nu \rightarrow -(CHCHCH_2CH)- + H_2O + 2 \cdot Cl \qquad (11)$$

$$-CH \cdot CHCH_2CH \cdot CH_2- + 2 \cdot Cl \rightarrow -CHCH = CHCH = CH- + 2 HCl \qquad (12)$$

$$-CH = CH\text{-}CH_2- + {}^1O_2 \rightarrow -(HOO)CH\text{-}CH = CH- \qquad (13)$$

$$-(CHCH = CHCH)- + O_2 \rightarrow \rightarrow CO_2 + H_2O \qquad (14)$$

## 4. Conclusions

An $La^{3+}/TiO_2$ photocatalyst was successfully prepared and characterized by the ball milling method. The characterization results showed that $La^{3+}$ successfully doped modified $TiO_2$. $La^{3+}$ was dispersed on the surface of $TiO_2$, increasing the surface oxygen vacancies and lattice distortion. Compared with $TiO_2$, the $La^{3+}/TiO_2$ photocatalyst showed higher photocatalytic activity. $La^{3+}/TiO_2$ was applied to photocatalytic degradation of PVC plastic. After 30 h of UV light irradiation, the weight loss rate of the PVC and $TiO_2/PVC$ were 2.12% and 8.94%, respectively, while that of the $La^{3+}/TiO_2/PVC$ ($La^{3+}/TiO_2$ 1.5 wt%) reached 17.78%, which was eight times the degradation rate of PVC and two times the degradation rate of $TiO_2/PVC$. The $La^{3+}/TiO_2/PVC$ composite film showed higher photodegradability. After illumination, the FT-IR spectrum showed a new peak at 1720 cm$^{-1}$, which belonged to C = O, and the XRD patterns showed a new peak at 32.5°. The surface pore size of the $La^{3+}/TiO_2/PVC$ sample increased to 2~8 μm. These results indicated that the $La^{3+}/TiO_2/PVC$ film had photodegradability. Predictably, $La^{3+}/TiO_2/PVC$ has a good application prospect, and it is an environment-friendly composite material.

**Author Contributions:** Conceptualization, Z.S. and T.S.; methodology, D.Z.; software, Y.Z.; validation, C.L. and J.L.; formal analysis, B.L.; investigation, Y.Z.; resources, Z.S.; data curation, Y.Z.; writing—original draft preparation, Y.Z.; writing—review and editing, T.S.; supervision, T.S.; project administration, Z.S.; funding acquisition, Z.S. All authors have read and agreed to the published version of the manuscript.

**Funding:** This research received no external funding.

**Institutional Review Board Statement:** Not applicable.

**Informed Consent Statement:** Not applicable.

**Data Availability Statement:** Not applicable.

**Acknowledgments:** This research was supported by the National Natural Science Foundation of China (Grant No. 22168017), the Hainan Provincial Natural Science Foundation of China (2019RC183, 2019RC187, 420QN259, 222CXTD513 and 420QN251), and the Scientific Research Project of Higher Education of Hainan Province (HNKY 2020–2026), the Graduate Innovative Research Project of Hainan Province (Qhyb2022-104).

**Conflicts of Interest:** The authors declare no conflict of interest.

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
