# Peer review of "Ball-Milling Preparation of La3+/TiO2 Photocatalyst and Application in Photodegradation of PVC Plastics"

_coatings, doi:10.3390/coatings13020317_

Round 1

Reviewer 1 Report

 In this article, La3+ modified TiO2 nano-particles were prepared by ball milling and characterized. La3+/TiO2 was mixed with Polyviny chloride (PVC) plastic to make a photodegradable composite film, and the photodegradation performance and mechanical properties of films were evaluated. After 30 h UV irradiation, the weight loss rate of the PVC and TiO2/PVC were only 2.12% and 8.94%, respectively, while that of the La3+/TiO2/PVC film reached 17.78%, which was 8 times the degradation rate of PVC and 2 times the degradation rate of TiO2/PVC.

Although this work seems interesting, but before publishing in this journal, I recommend some major changes as given below.

1-      I recommend the authors to add some more details about this work in the abstract section.

2-      The Legends/Captions of Figure 1 need to be rewrite. Please write the captions in details which will provide a suitable information for the readers.

3-      Some of the Figures presented in this review articles are not clearly readable e.g., Figure 5. I suggest the authors to add the high quality or readable Figures.

4-      I strongly recommend the author to cite these recently reported articles on sensors in the introduction section, ( https://doi.org/10.1002/adfm.202204781),  (https://doi.org/10.1016/j.jallcom.2022.165815) and (https://doi.org/10.1002/advs.202204779).

5-      There are some typo errors and repeating terms/ word and also  English language need to be improved.

Remarks: Publish after Major revision

Author Response

Dear Editor and Reviewers,   

Thank you very much for your helpful comments and suggestions on our manuscript (Manuscript ID: coatings-2120354). We have checked and revised our manuscript carefully according to the comments. The revised places have been highlighted in the text. Our responses to the comments are listed as follows. In this paper, the La3+ modified TiO2 nanoparticles were prepared by the ball milling method and characterized. La3+/TiO2 was mixed with Polyvinyl chloride (PVC) plastic to make a photodegradable composite film, and the photodegradation and mechanical properties of the composite films were evaluated. The photocatalytic technology is a feasible and effective way to solve "white pollution". Hence, we believe that this paper is appropriate for “coatings”, and it would be very grateful if this manuscript could be considered for publication after this revision. Thanks again.

Best wishes!

Sincerely yours,
Associate Professor . Tianyi Sun

School of Chemistry and Chemical Engineering, Hainan Normal University, Haikou 571127, China

Key Laboratory of Water Pollution Treatment and Resource Reuse of Hainan Province, Haikou 571127, China.

Responses to the Comments

Response to Reviewer #1:

1.I recommend the authors to add some more details about this work in the abstract section.

Response: Thanks for your suggestion. More details have been added in the abstract section on the page 1 and marked in yellow.

2.The Legends/Captions of Figure 1 need to be rewrite. Please write the captions in details which will provide a suitable information for the readers.

Response: Thanks for your suggestion. The Legends/Captions of Figure 1 have been revised to provide the suitable information on the page 5, the section 2.5.

3.Some of the Figures presented in this review articles are not clearly readable e.g., Figure 5. I suggest the authors to add the high quality or readable Figures.

Response: Thanks for your suggestion. To make Figure 2 and Figure 5 clearer, the font size has been changed, and the high quality picture has been replaced on page 6 and 9.

4.I strongly recommend the author to cite these recently reported articles on sensors in the introduction section, ( https://doi.org/10.1002/adfm.202204781),  (https://doi.org/10.1016/j.jallcom.2022.165815)and(https://doi.org/10.1002/advs.202204779).

Response: Thanks for your suggestion. These recently reported articles have been cited and added in the introduction on page 2.

References are as follows:

[10] Dastgeer, G.; Shahzad, Z.M.; Chae, H.; Kim, Y.H.; Ko, B.M.; Eom, J. Bipolar junction transistor exhibiting excellent output characteristics with a prompt response against the selective protein. Adv. Func.t Mater. 2022, 32, 2204781.

[11] Dastgeer, G.; Afzal, A.M.; Jaffery, S. H.A.; Imran, M.; Assiri, M.A.; Nisar, S. Gate modulation of the spin current in graphene/WSe2 van der Waals heterostructure at room temperature. J. Alloy. Compd. 2022, 919, 919.

[13] Dastgeer, G.; Nisar, S.; Shahzad, Z.M.; Rasheed, A.; Kim, D.; Jaffery, S.H.A.; Wang, L.; Usman, M.; Eom, J. Low‐power negative‐differential‐resistance device for sensing the selective protein via supporter molecule engineering. Adv. Sci. 2022, 2204779.

5.There are some typo errors and repeating terms/word and also  English language need to be improved.

Response: Thanks for your suggestion. The manuscript has been checked and revised. The revised places have been highlighted in the text.

Reviewer 2 Report

The paper is worth to be proceed, and the problem described in it is interesting. I have some suggestions to improve the paper before publishing:

In table 1 the are some results of the mechanical properties of the mentioned films but in the previous part of the paper, there is no information about methods of examining such properties.

The photocatalytic degradation mechanism which is shown in Fig 10 should be better explained in the text

Formula 1 should be written in maths format. 

There is no reference from the past two years. A wider literature study should be done. 

Author Response

Response to Reviewer #2:

1.In table 1 the are some results of the mechanical properties of the mentioned films but in the previous part of the paper, there is no information about methods of examining such properties.

Response: Thanks for your suggestion. A detailed explanation has been added on the page 4 and marked in yellow, in the section 2.2. The details are as follows:

The film was cut into a rectangle of 50 mm×10 mm and fixed on the digital tensile testing machine with an initial spacing of 20 mm, running at a 50 mm/min. The thickness of the film was measured 5 times with the NSCING electronic digital micrometer, and the average value was taken.

2.The photocatalytic degradation mechanism which is shown in Fig 10 should be better explained in the text

Response: Thanks for your suggestion. The photocatalytic degradation mechanism shown in Fig. 10 has been explained in more detail, and the formulas 2-4 have been added on page 13-14.

3.Formula 1 should be written in maths format. 

Response: Thanks for your suggestion. Formula 1 has been written and modified using Formula Editor on Page 5.

4.There is no reference from the past two years. A wider literature study should be done. 

Response: Thanks for your suggestion. The literature in recent years has been cited and replaced.

References are as follows:

[8] Zhang, Y.; Sun, T.; Zhang, D.; Shi, Z.; Zhang, X.; Li, C.; Wang, L.; Song, J.; Lin, Q. Enhanced photodegradability of PVC plastics film by codoping nano-graphite and TiO2. Polym. Degrad. Stabil. 2020, 181, 109332.

[10] Dastgeer, G.; Shahzad, Z.M.; Chae, H.; Kim, Y.H.; Ko, B.M.; Eom, J. Bipolar junction transistor exhibiting excellent output characteristics with a prompt response against the selective protein. Adv. Func.t Mater. 2022, 32, 2204781.

[11] Dastgeer, G.; Afzal, A.M.; Jaffery, S. H.A.; Imran, M.; Assiri, M.A.; Nisar, S. Gate modulation of the spin current in graphene/WSe2 van der Waals heterostructure at room temperature. J. Alloy. Compd. 2022, 919, 919.

[13] Dastgeer, G.; Nisar, S.; Shahzad, Z.M.; Rasheed, A.; Kim, D.; Jaffery, S.H.A.; Wang, L.; Usman, M.; Eom, J. Low‐power negative‐differential‐resistance device for sensing the selective protein via supporter molecule engineering. Adv. Sci. 2022, 2204779.

[14] Tsebriienko, T.; Popov, A.I. Effect of poly (titanium oxide) on the viscoelastic and thermophysical properties of interpenetrating polymer networks. Crystals. 2021, 11, 794.

[15] Serga,V.; Burve, R.; Krumina, A.; Romanova, M.; Kotomin, E.A.; Popov, A.I. Extraction–pyrolytic method for TiO2 polymorphs production. Crystals, 2021,11, 431.

[16] Serga, V.; Burve, R.; Krumina, A.; Pankratova, V.; Popov, A.; Pankratov, V. Study of phase composition, photocatalytic activity, and photoluminescence of TiO2 with Eu additive produced by the extraction-pyrolytic method. J. Mater. Res. Technol. 2021, 13, 2350-2360.

[17] Cerrato, E.; Gaggero, E.; Calza, P.; Paganini, M.C. The role of Cerium, Europium and Erbium doped TiO2 photocatalysts in water treatment: a mini-review. Adv. Chem. Eng. 2022, 10, 100268.

Reviewer 3 Report

This article can be recommended for published after the necessary revision and corresponding improvements:

1.     More updated information about the TiO2,  its modifications and especially effects of different dopants will be useful to give in the Introduction. See , for example:

Serga, V., Burve, R., Krumina, A., et al (2021). Study of phase composition, photocatalytic activity, and photoluminescence of TiO2 with Eu additive produced by the extraction-pyrolytic method. journal of materials research and technology13, 2350-2360.

Cerrato, E., Gaggero, E., Calza, P., & Paganini, M. C. (2022). The role of Cerium, Europium and Erbium doped TiO2 photocatalysts in water treatment: a mini-review. Chemical Engineering Journal Advances, 100268.

2.     A short introduction about polymer-oxide composites is also desirable. New specific effects in this case have already been stated in the case of polymer-Fe3O4 and polymer-BaZrO3. See:   Aksimentyeva, O. I., et al (2014). Modification of polymer-magnetic nanoparticles by luminescent and conducting substances. Molecular Crystals and Liquid Crystals590(1), 35-42. https://doi.org/10.1080/15421406.2013.873646

3.     Did the authors try to analyze the additional absorption (4b) by analyzing it, decomposing it into Gaussians and recalculating the horizontal axis in energy units (eV).

4.     Whether a change in such spectra was observed if they were measured during the aging of samples ?

Author Response

Response to Reviewer #3:

1.More updated information about the TiO2,  its modifications and especially effects of different dopants will be useful to give in the Introduction. See, for example:

Serga, V., Burve, R., Krumina, A., et al (2021). Study of phase composition, photocatalytic activity, and photoluminescence of TiO2 with Eu additive produced by the extraction-pyrolytic method. journal of materials research and technology, 13, 2350-2360.

Cerrato, E., Gaggero, E., Calza, P., & Paganini, M. C. (2022). The role of Cerium, Europium and Erbium doped TiO2 photocatalysts in water treatment: a mini-review. Chemical Engineering Journal Advances, 100268.

Response: Thanks for your suggestion. More updated information about the TiO2 has been provided and added on page 2, in the introduction.

References are as follows:

[16] Serga, V.; Burve, R.; Krumina, A.; Pankratova, V.; Popov, A.; Pankratov, V. Study of phase composition, photocatalytic activity, and photoluminescence of TiO2 with Eu additive produced by the extraction-pyrolytic method. J. Mater. Res. Technol. 2021, 13, 2350-2360.

[17] Cerrato, E.; Gaggero, E.; Calza, P.; Paganini, M.C. The role of Cerium, Europium and Erbium doped TiO2 photocatalysts in water treatment: a mini-review. Adv. Chem. Eng. 2022, 10, 100268.

2.A short introduction about polymer-oxide composites is also desirable. New specific effects in this case have already been stated in the case of polymer-Fe3O4 and polymer-BaZrO3. See: Aksimentyeva, O. I., et al (2014). Modification of polymer-magnetic nanoparticles by luminescent and conducting substances. Molecular Crystals and Liquid Crystals, 590(1), 35-42. https://doi.org/10.1080/15421406.2013.873646

Response: Thanks for your suggestion. As Reviewer say, this article is suitable for the introduction of composite materials, which has been quoted and added on page 2.

References are as follows:

[19] Aksimentyeva, O.I.; Savchyn, V.P.; Dyakonov, V.P.; Piechota, S.; Y.Y.; Horbenko,  Opainych, I.Y.; Demchenko, P.Y.; Popov, A.; Szymczak, H. Modification of polymer-magnetic nanoparticles by luminescent and conducting substances. Mol. Cryst. Liq. Cryst. 2014, 590 , 35-42.

3.Did the authors try to analyze the additional absorption (4b) by analyzing it, decomposing it into Gaussians and recalculating the horizontal axis in energy units (eV).

Response: We appreciate the reviewer ' s insightful suggestion and agree that it would be useful for analysis. However, the theoretical knowledge analysis of Gaussians is lacking at present, we will conduct in-depth research and analysis on this part of work in the future.

4.Whether a change in such spectra was observed if they were measured during the aging of samples ?

Response: Thanks for your suggestion. The supplementary test by the FT-IR spectra (Figure 6B) confirmed that the film had obvious changes in the process of photodegradation. The explanation and Figure 6B have been added on the page 9-10. The details are as follows:

The intensity of the characteristic peak of C=O structure of the La3+/TiO2/PVC film increased gradually after 0 h,10 h, 20 h and 30 h of irradiation (Figure 6B), indicating that the film was degraded with the continuous irradiation of light source.

Reviewer 4 Report

Referee Report on manuscript “Ball-milling Preparation of La3+/TiO2 Photocatalyst and Application in Photodegradation of PVC Plastics”.

This is quite a good article, which can be published after the necessary revision and improvement:

1.     More introductory information about the TiO2 under consideration must be given. Titanium oxide has several modifications and each of them works in its own way. In addition, these effects depend on the size of the nanoparticles. This remains outside the introduction and is slightly misunderstood. This important part is supported by two references [14, 15], both are quite old.   There are a lot recently published MDPI paper on similar subject, which can be mentioned here:

Tsebriienko, T., & Popov, A. I. (2021). Effect of poly (titanium oxide) on the viscoelastic and thermophysical properties of interpenetrating polymer networks. Crystals11(7), 794

 Serga, V., Burve, R., Krumina, A., et al (2021). Extraction–pyrolytic method for TiO2 polymorphs production. Crystals11(4), 431.

 Soundarya, T. L., Jayalakshmi, T., Alsaiari, M. A., et al (2022). Ionic Liquid-Aided Synthesis of Anatase TiO2 Nanoparticles: Photocatalytic Water Splitting and Electrochemical Applications. Crystals12(8), 1133.

and many others.

2.     A deeper interpretation of the absorption spectra (Fig.4 and B) is required using the appropriate literature analysis.

3.     Figure 5 has very insufficient quality and needs improvement.

4.     Figure 5(a).  In general, the time on the horizontal axis depends on the intensity of the light and also on its spectral composition. Is it possible to specify other units instead of time, for example, photons/cm2 ?

5.     Lines 181-183. This sentence looks unfounded and therefore requires physical justification.

6.     It would be useful to summarize all the peaks in Figure 6 in one table and, together with the literature data, give a comprehensive interpretation.

7.     Can you clearly formulate in your conclusions what new data on these materials were obtained in this work.

 In principle, the article is interesting and can be recommended for publication after due consideration of the above comments.

Author Response

Response to Reviewer #4:

1.More introductory information about the TiO2 under consideration must be given. Titanium oxide has several modifications and each of them works in its own way. In addition, these effects depend on the size of the nanoparticles. This remains outside the introduction and is slightly misunderstood. This important part is supported by two references [14, 15], both are quite old.   There are a lot recently published MDPI paper on similar subject, which can be mentioned here:

Tsebriienko, T., & Popov, A. I. (2021). Effect of poly (titanium oxide) on the viscoelastic and thermophysical properties of interpenetrating polymer networks. Crystals, 11(7), 794;

 Serga, V., Burve, R., Krumina, A., et al (2021). Extraction–pyrolytic method for TiO2 polymorphs production. Crystals, 11(4), 431; 

Soundarya, T. L., Jayalakshmi, T., Alsaiari, M. A., et al (2022). Ionic Liquid-Aided Synthesis of Anatase TiO2 Nanoparticles: Photocatalytic Water Splitting and Electrochemical Applications. Crystals, 12(8), 1133.

Response: Thanks for your suggestion. More introductory information about the TiO2 have been provided and added in the introduction on page 2. In addition, the obove original references have been also cited.

References are as follows:

[14] Tsebriienko, T.; Popov, A.I. Effect of poly (titanium oxide) on the viscoelastic and thermophysical properties of interpenetrating polymer networks. Crystals. 2021, 11, 794.

[15] Serga,V.; Burve, R.; Krumina, A.; Romanova, M.; Kotomin, E.A.; Popov, A.I. Extraction–pyrolytic method for TiO2 polymorphs production. Crystals2021,11, 431.

[18] Soundarya, T.L.; Jayalakshmi, T.; Alsaiari, M.A.; Jalalah, M.; Abate, A.; Alharthi, F.A.; Ahmad, N.; Nagaraju, G. Ionic liquid-aided synthesis of anatase TiO2 nanoparticles: photocatalytic water splitting and electrochemical applications. Crystals, 2022, 12, 1133.

2.A deeper interpretation of the absorption spectra (Fig.4 and B) is required using the appropriate literature analysis.

Response: Thanks for your suggestion. Figure 4 has been explained in more depth and quoted in appropriate literature, which has been added on page 7.

References are as follows:

[31] Liu, R.; Ren, Y.; Wang, J.; Wang, Y.; Jia, J.; Zhao, G. Preparation of Nb-doped TiO2 films by sol-gel method and their dual-band electrochromic properties. Ceram. Int. 2021, 47, 31834-31842.

[32] Xu, Z.; Jiang, Y.; Gui, C. Based on the Study on Preparation and Properties of Co (II) and La3+-Doped TiO2 Electrochromic Film. Integr. Ferroelectr. 2020, 209, 58-67.

3.Figure 5 has very insufficient quality and needs improvement.

Response: Thanks for your suggestion. In order to display it more clearly, Figure 5 has been modified on page 9.

4.Figure 5(a).  In general, the time on the horizontal axis depends on the intensity of the light and also on its spectral composition. Is it possible to specify other units instead of time, for example, photons/cm2 ?

Response: Thanks for your suggestion. In this experiment, the 300 W medium-pressure ultraviolet light source with main wavelength of 365 nm was used to irradiate, and the distance between the light source and the sample was 20 cm. The sample was taken out every 5 h, and the total illumination time was 30 h. Under the same light intensity, the weight change of sample was monitored as the illumination time changed, so the horizontal axis was more appropriate for time change.

5.Lines 181-183. This sentence looks unfounded and therefore requires physical justification.

Response: Thanks for your suggestion. Lines 181-183. This sentence has been reported in relevant literature, and the relevant literature has been added.

Reference was as follow:

[39] Chakrabarti, S.; Chaudhuri, B.; Bhattacharjee, S.; Das, P.; Dutta, B.K. degradation mechanism and kinetic model for photocatalytic oxidation of PVC–ZnO composite film in presence of a sensitizing dye and uv radiation. J. Hazard. Mater. 2008, 154, 230-236.

6.It would be useful to summarize all the peaks in Figure 6 in one table and, together with the literature data, give a comprehensive interpretation.

Response: Thanks for your suggestion. The characteristic peaks in Figure 6 have been compiled into Table 1, and a more detailed description has been added on page 9-10.

Table 1 The characteristic peak of infrared spectrum structure

Serial number

Wavenumber/cm-1

Characteristic peak

1

2973, 2917, 1427

C-H

2

698, 624

C-Cl

3

1097

C-C

4

478

Ti-O-Ti

5

1720

C=O

7.Can you clearly formulate in your conclusions what new data on these materials were obtained in this work.

Response: Thanks for your suggestion. The new data obtained from this work have been supplemented in the conclusion, which has been added on page15.

Round 2

Reviewer 1 Report

Accept it in the present form.

Author Response

Dear Editor,   

Thank you very much for your helpful comments and suggestions on our manuscript (Manuscript ID: coatings-2120354). We have removed two citation corresponding to Reviewer 1. The revision of the article used the function of "tracking changes" of Microsoft Word. Thank you very much for your acceptance of this article.

Best wishes!

Sincerely yours,
Associate Professor . Tianyi Sun

School of Chemistry and Chemical Engineering, Hainan Normal University, Haikou 571127, China

Key Laboratory of Water Pollution Treatment and Resource Reuse of Hainan Province, Haikou 571127, China.

Reviewer 2 Report

In my opinion, after improvement, the paper is ready to be published

Author Response

(The authors gave the same response as above.)

Reviewer 3 Report

After successful revision, this paper can be accepted

Author Response

(The authors gave the same response as above.)

Reviewer 4 Report

After very constructive revision this manuscript can be recommended for publication

Author Response

(The authors gave the same response as above.)
